# BLOCKWISE SELF-ATTENTION FOR LONG DOCUMENT UNDERSTANDING

## ABSTRACT

We present BlockBERT, a lightweight and efficient BERT model that is designed to better modeling long-distance dependencies. Our model extends BERT by introducing sparse block structures into the attention matrix to reduce both memory consumption and training time, which also enables attention heads to capture either short- or long-range contextual information. We conduct experiments on several benchmark question answering datasets with various paragraph lengths. Results show that BlockBERT uses 18.7-36.1% less memory and reduces the training time by 12.0-25.1%, while having comparable and sometimes better prediction accuracy, compared to an advanced BERT-based model, RoBERTa.

## 1 INTRODUCTION

Recent emergence of the *pre-training* and *fine-tuning* paradigm, exemplified by methods like ELMo (Peters et al., 2018), GPT-2 (Radford et al., 2019), BERT (Devlin et al., 2019), XLNet (Yang et al., 2019) and RoBERTa (Liu et al., 2019), has drastically reshaped the landscape of the natural language processing research. These methods first pre-train a deep model with language model objectives using a large corpus and then fine-tune the model using in-domain supervised data for target applications. Despite its conceptual simplicity, this paradigm has reestablished the new state-of-the-art baselines across various tasks, such as question answering (Devlin et al., 2019), coreference resolution (Joshi et al., 2019b), relation extraction (Soares et al., 2019) and text retrieval (Lee et al., 2019; Nogueira & Cho, 2019), to name a few.

Building such models in practice, however, is an extremely resource-intensive process. For instance, the training of BERT-family models is notoriously expensive. Devlin et al. (2019) report that it takes four days for pre-training BERT-Base/BERT-Large on 4/16 Cloud TPUs, respectively. In order to reduce the pre-training time of RoBERTa to 1 day, Liu et al. (2019) use 1,024 V100 GPUs. One crucial factor that contributes to the long training time is the memory consumption of these deep models, as it directly affects the batch size. Although the fine-tuning stage is relatively inexpensive, the memory issue still restricts the scenarios in which BERT can be used. For instance, "it is currently not possible to re-produce most of the BERT-Large results on the paper using a GPU with 12GB-16GB of RAM, because the maximum batch size that can fit in memory is too small.[1]"

Although one may think that model size is the main contributor to the large memory consumption, our analysis (Section 2.1) shows that one of the main bottlenecks is actually dot-product self-attention, operated in multiple layers of Transformers (Vaswani et al., 2017), the building block of BERT. As the attention operation is quadratic to the sequence length, this fundamentally limits the maximum length of the input sequence, and thus restricts the model capacity in terms of capturing long-distance dependencies. As a result, downstream tasks have to either truncate their sequences to leading tokens (Nogueira & Cho, 2019) or split their sequences with a sliding window (Joshi et al., 2019a;b). Ad-hoc handling of long sequences is also required in the pre-training stage, such as updating the model using only short sequences in the early stage (Devlin et al., 2019).

Common strategies for reducing memory consumption, unfortunately, do not work. For instance, shrinking the model by lowering the number of layers $L$, attention heads $A$, or hidden units $H$ leads to significant performance degradation (Vaswani et al., 2017; Devlin et al., 2019) and does not address the long sequence issue. Alternatively, general low-memory training techniques, such as

---

[1]https://github.com/google-research/bert

microbatching (Huang et al., 2018) and gradient checkpointing (Chen et al., 2016) essentially trade off training time for memory consumption, prolongs the already lengthy training process.

In this work, we explore a different strategy, *sparsifying the attention layers*, intending to design a lightweight and effective BERT that can model long sequences in a memory-efficient way. Our BlockBERT extends BERT by introducing sparse block substructures into the attention matrix to reduce both memory consumption and the number of floating point operations (FLOPs), which also enables attention heads to capture either short- or long-range contextual information. Compared to the previous method that also enforces sparsity (e.g., Child et al. (2019)), our approach is much simpler mathematically and very easy to implement. More importantly, the results of experiments conducted on several benchmark question answering datasets with various paragraph lengths show that BlockBERT performs comparably or even better than the original BERT-family models, while enjoying an 18.7-36.1% reduction in memory usage and 12.0-25.1% reduction in training time.

The rest of the paper is organized as follows. Section 2 gives a brief introduction of the BERT model, along with an in-depth analysis of its memory usage during training time. We describe our proposed model in Section 3 and contrast it with existing methods that aim for creating a lighter model. Section 4 presents the experimental results and ablation studies, followed by a short survey of other related work in Section 5 and the conclusion in Section 6.

## 2 BACKGROUND: MEMORY BOTTLENECK IN TRAINING BERT

We briefly review BERT and introduce its memory profiling in this section. Following the paradigm of language model pre-training and down-stream task fine-tuning, BERT (Devlin et al., 2019) consists of multiple layers of bidirectional Transformers (Vaswani et al., 2017), where each Transformer encoder has a multi-head self-attention layer and a position-wise feed-forward layer. Using the same notation as in (Devlin et al., 2019), we denote the number of Transformer layers by $L$, the number of hidden units by $H$, the number of attention heads by $A$, the sequence length by $N$ and the batch size by $B$. We also assume the feed-forward hidden unit size to be $4H$.[2]

### 2.1 MEMORY PROFILING

Training BERT is a memory-intensive process. In order to identify the bottleneck, we follow the memory model proposed by Sohoni et al. (2019), where the memory usage throughout neural network training is categorized into three main types: (1) **Model Memory** is used to store model parameters; (2) **Optimizer Memory** is the additional memory used by the specific learning algorithm during the process; (3) **Activation Memory** consists of the outputs of each layer, which are cached for reuse in backpropagation to compute gradients.

Take BERT-Base training as an example. The model has 110M parameters, so the model memory uses 0.2 GB if stored in half-precision floating-point format (FP16). For Adam (Kingma & Ba, 2014), the optimizer needs additional memory to store the gradients, first moments, and second moments of model parameters. If stored using the same precision, the optimizer memory should be three times of model memory.[3] To calculate the exact size of activation memory is not trivial because it depends heavily on the implementation of the toolkit. Instead, we measure it empirically by training BERT-Base using Adam with a memory profiler (more details are provided in Appendix A.2).

We use 32 NVIDIA V100 GPUs for training. Each single GPU thus consumes a mini-batch of size $b = B/32 = 8$. Figure 1a shows the profiling result for a single GPU, where the model/optimizer/activation memory consumes 0.21/1.03/8.49 GB, resp. We can see that activation memory accounts for the vast majority of the total GPU memory (87.6%) and is clearly the bottleneck. Notice that although our analysis is done on BERT-Base, it can be easily generalized to BERT-Large and other models such as RoBERTa (Liu et al., 2019) and XLNet (Yang et al., 2019).

---

[2]The default parameter settings for BERT-Base and BERT-Large can be found in Table 5 in Appendix A.1.

[3]In the current PyTorch Adam implementation, the first and second moments are stored in single precision. Consequently, BERT's optimizer memory (1 GB) is five times of model memory (0.2 GB).

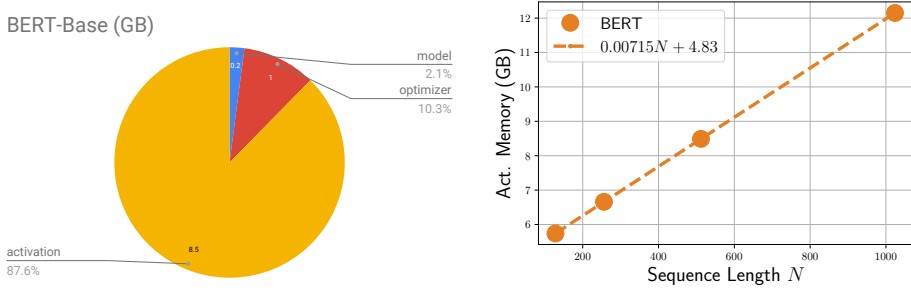

(a) BERT-Base Training Memory Profiling      (b) Regression Analysis on Activation Memory

Figure 1: Memory Profiling for BERT

## 2.2 A REGRESSION ANALYSIS ON ACTIVATION MEMORY

For BERT, or more specifically, Transformer, the activation memory corresponds to intermediate results of different layers It grows linearly in all the model hyper-parameters, except the sequence length $N$, due to the attention layers. To quantify more clearly the $O(N)$ and $O(N^2)$ components in the activation memory, we conduct a regression analysis as follows. Assume that the activation memory (in each GPU) is a polynomial $a_2 b N^2 + a_1 b N + a_0$, where $b$ is the batch size in each GPU. If we fix the total number of tokens in a GPU, i.e., $b \times N$, to be constant (in our case, 4096), we should have a linear function w.r.t. $N$, i.e., $4096 a_2 N + 4096 a_1 + a_0$. We enumerate $N$ from $\{128, 256, 512, 1024\}$ in our experiments, and plot the corresponding profiled activation memory in Figure 1b. Using ordinary least squares (OLS), with $b \times N = 4096$, the estimated linear function for activation memory is $0.00715 \times N + 4.83$, where the first term is responsible for the $O(N^2)$ component. When $N = 512$, we can see that for BERT-Base, the $O(N^2)$ component accounts for 3.66 GB and $O(N)$ accounts for 4.83 GB. When the sequence length $N$ increases to 1024, however, the $O(N^2)$ component increases to 7.32 GB, while $O(N)$ is unchanged.

## 2.3 GENERAL TECHNIQUES FOR REDUCING MEMORY USAGE IN MODEL TRAINING

Observing that activation memory is the bottleneck, we discuss the effectiveness of common memory reduction techniques for BERT training below.

**Low Precision** (Micikevicius et al., 2017): Low precision is to use half-precision or mixed-precision for training neural networks. This technique has been widely used in Transformer training (Ott et al., 2019; Liu et al., 2019). In this work, we already assume to use mixed-precision training by default, as indicated in the aforementioned analysis.

**Microbatching** (Huang et al., 2018): Microbatching is to split a batch into small micro-batches (which can be fit into memory), and then run forward and backward passes on them separately with gradients for each micro-batch accumulated. Because it runs forward/backward pass multiple times for a single batch, it trades off time for memory.

**Gradient Checkpointing** (Chen et al., 2016): Gradient checkpointing saves memory by only caching activations of a subset of layers. The un-cached activations will be recomputed during backpropagation from the latest checkpoint. This strategy trades off time for memory by repeating computations that require large memory and will obviously extend the model training time.

**Knowledge Distillation** (Hinton et al., 2015): Knowledge distillation aims to compress and transfer knowledge from a teacher model to a simpler student model. However, knowledge distillation relies on a teacher model (which is still expensive in training time) and usually suffers from a certain degree of performance degradation.

As common techniques are limited in reducing both the training time and memory usage, we investigate how to optimize the dot-product attention layers and introduce our approach next.

## 3 MODEL: BLOCKBERT

Following (Vaswani et al., 2017), the dot-product attention in Transformer is defined as:

$$\text{Attention}(\boldsymbol{Q}, \boldsymbol{K}, \boldsymbol{V}) = \text{softmax}\left(\frac{\boldsymbol{Q}\boldsymbol{K}^\top}{\sqrt{d}}\right)\boldsymbol{V}.$$

where $\boldsymbol{Q}, \boldsymbol{K}, \boldsymbol{V} \in \mathbb{R}^{N \times d}$ with $N$ to be the sequence length and $d$ to be a hidden dimension. As we can see, the inner product between $\boldsymbol{Q}$ and $\boldsymbol{K}$ consumes $O(N^2)$ memory. One simple way to reduce memory consumption of attention is to sparsify the attention matrix. Suppose we have a masking matrix $\boldsymbol{M} \in \{0, 1\}^{N \times N}$. We define a masked version of attention as follows:

$$\text{Attention}(\boldsymbol{Q}, \boldsymbol{K}, \boldsymbol{V}, \boldsymbol{M}) = \text{softmax}\left(\frac{\boldsymbol{Q}\boldsymbol{K}^\top}{\sqrt{d}} \odot \boldsymbol{M}\right)\boldsymbol{V}, \tag{1}$$

with operator $\odot$ defined by

$$(\boldsymbol{A} \odot \boldsymbol{M})_{ij} = \begin{cases} \boldsymbol{A}_{ij} & \text{if } \boldsymbol{M}_{ij} = 1 \\ -\infty & \text{if } \boldsymbol{M}_{ij} = 0 \end{cases}.$$

In this work, we design $\boldsymbol{M}$ to be a *sparse block matrix*, which not only reduces memory and the number of floating point operations (FLOPs) but also benefits from efficient dense matrix support from deep learning frameworks, such as PyTorch and Tensorflow. More formally, we split the length-$N$ input sequence into $n$ partitions, with each partition of length $\frac{N}{n}$.[4] The $N \times N$ attention matrix is then partitioned into $n \times n$ blocks, where each block matrix is of size $\frac{N}{n} \times \frac{N}{n}$. A sparse block matrix $\boldsymbol{M}$ can be defined by a permutation $\pi$ of $\{1, 2, \cdots, n\}$:

$$\boldsymbol{M}_{ij} = \begin{cases} 1 & \text{if } \pi\left(\lfloor \frac{(i-1)n}{N} + 1 \rfloor\right) = \lfloor \frac{(j-1)n}{N} + 1 \rfloor \\ 0 & \text{otherwise.} \end{cases} \tag{2}$$

By writing $\boldsymbol{Q}, \boldsymbol{K}, \boldsymbol{V}$ as be block matrices, such that $\boldsymbol{Q} = \begin{bmatrix} \boldsymbol{Q}_1^\top & \cdots & \boldsymbol{Q}_n^\top \end{bmatrix}^\top$, $\boldsymbol{K} = \begin{bmatrix} \boldsymbol{K}_1^\top & \cdots & \boldsymbol{K}_n^\top \end{bmatrix}^\top$ and $\boldsymbol{V} = \begin{bmatrix} \boldsymbol{V}_1^\top & \cdots & \boldsymbol{V}_n^\top \end{bmatrix}^\top$ and pluging them into Equation 1, we can formally define Blockwise Attention as follows:

$$\text{Blockwise-Attention}(\boldsymbol{Q}, \boldsymbol{K}, \boldsymbol{V}, \boldsymbol{M}) = \begin{bmatrix} \text{softmax}\left(\frac{\boldsymbol{Q}_1 \boldsymbol{K}_{\pi(1)}^\top}{\sqrt{d}}\right)\boldsymbol{V}_{\pi(1)} \\ \vdots \\ \text{softmax}\left(\frac{\boldsymbol{Q}_n \boldsymbol{K}_{\pi(n)}^\top}{\sqrt{d}}\right)\boldsymbol{V}_{\pi(n)} \end{bmatrix}. \tag{3}$$

As a result, it only needs to compute and store $\boldsymbol{Q}_i \boldsymbol{K}_{\pi(i)}^\top$ $(i = 1, \cdots n)$, each of which has size $\frac{N}{n} \times \frac{N}{n}$. In other words, BLOCKBERT reduces the corresponding $O(N^2)$ memory consumption and FLOPs by a factor of $n$, since $\frac{N}{n} \times \frac{N}{n} \times n = \frac{N \times N}{n}$.

### 3.1 BLOCKWISE MULTI-HEAD ATTENTION

Analogous to Multi-head Attention (Vaswani et al., 2017), we allow queries, keys, and values to be projected multiple times and perform blockwise attentions in parallel. Moreover, different block-wise attention heads can use different masking matrices. The outputs of multiple heads are then concatenated and aggregated with another linear projection. Let $A$ be the number of attention heads and $H$ the number of hidden units. *Blockwise multi-head attention* is formally defined by:

$$\text{Blockwise-Multi-head-Attention}(\boldsymbol{Q}, \boldsymbol{K}, \boldsymbol{V}) = \text{Concat}(\text{head}_1, \cdots \text{head}_A)\boldsymbol{W}^O$$

$$\text{where head}_i = \text{Blockwise-Attention}(\boldsymbol{Q}\boldsymbol{W}_i^Q, \boldsymbol{K}\boldsymbol{W}_i^K, \boldsymbol{V}\boldsymbol{W}_i^V, \boldsymbol{M}_i),$$

with $d = \frac{H}{A}$, $\boldsymbol{W}_i^Q, \boldsymbol{W}_i^K, \boldsymbol{W}_i^V \in \mathbb{R}^{H \times d}$ and the projection matrix $\boldsymbol{W}^O \in \mathbb{R}^{H \times H}$. Each masking matrix $\boldsymbol{M}_i$ is determined by permutation $\pi_i$ according to Equation 2. In particular, we choose $\pi$ from permutations generated by *shifting one position*: $\sigma = (2, 3, \cdots, n, 1)$, i.e., we select $\pi \in \{\sigma, \sigma^2, \cdots, \sigma^n\}$. For example, with 12 attention heads ($A = 12$) and 2 blocks ($n = 2$), one

---

[4]We assume $N$ can be divided by $n$. If not, we pad the input sequence to make $N$ divisible.

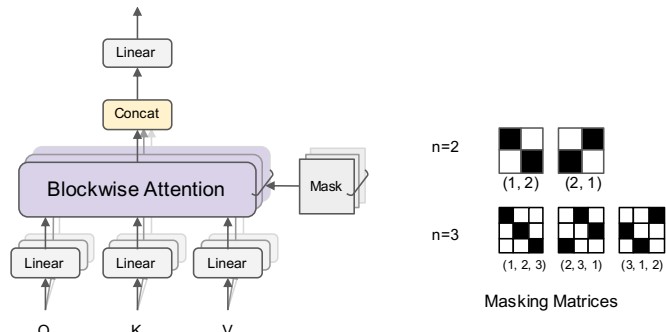

Figure 2: Architecture of Blockwise Multi-head Attention.

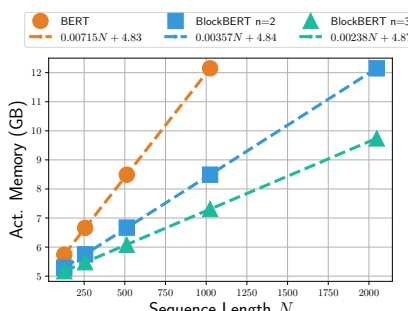

Figure 3: Regression analysis on activation memory for BERT and BlockBERT.

| $N$ | $b$ | Model | Act. Mem. (GB) | |
|---|---|---|---|---|
| | | | $O(N)$ | $O(N^2)$ |
| 512 | 8 | BERT | 4.83 | 3.66 |
| | | BlockBERT n=2 | 4.84 | 1.83 |
| | | BlockBERT n=3 | 4.87 | 1.22 |
| 1024 | 4 | BERT | 4.83 | 7.32 |
| | | BlockBERT n=2 | 4.84 | 3.66 |
| | | BlockBERT n=3 | 4.87 | 2.44 |

Table 1: Estimated $O(N^2)$ and $O(N)$ activation memory for BERT and BlockBERT.

configuration can be assigning 10 heads to permutation $(1, 2)$ and the other 2 heads to permutation $(2, 1)$. Figure 2 illustrates the blockwise multi-head attention with the block numbers $n \in \{2, 3\}$. Blockwise sparsity captures both local and long-distance dependencies in a memory-efficiency way, which is crucial for long-document understanding tasks. For instance, the identity permutation, i.e., $(1, 2, \cdots, n)$, enables each token to attend its nearby tokens in self-attention. Tokens within the same local group attend a long-distance group of tokens together in other permutations. Our proposed BlockBERT essentially replaces the multi-head attention layers in Transformer/BERT with blockwise multi-head attention.

## 3.2 ANALYSIS OF MEMORY USAGE REDUCTION

To validate our claim that BlockBERT with $n \times n$ blocks can reduce the $O(N^2)$ memory use by a factor of $n$, we perform the same memory profiling as described in sections 2.1 and 2.2. Again, We fix the number of tokens in each GPU ($b \times N = 4096$) and choose $N$ from $\{128, 256, 512, 1024, 2048\}$.[5] As we can see from Figure 3 and Table 1, the empirical results align well with the theoretical values. When we set block size to be 2 and 3 for BlockBERT, their estimated $O(N^2)$ activation memory decreases to 1/2 and 1/3 of BERT's $O(N^2)$ activation memory, resp. As shown in Table 2, for the sequence length $N = 512$, BlockBERT with 2 / 3 blocks saves 18.7% / 23.8% overall memory, resp. The saving is more significant for longer sequences. When $N = 1024$, the overall memory reduction of BlockBERT with 2 / 3 blocks is 27.3% / 36.1%, resp.

## 4 EXPERIMENTS

We evaluate the pre-training and fine-tuning performance of BlockBERT. In particular, when $n = 2$, we denote 10:2 to be the configuration which distributes 10 heads to permutation $(1, 2)$ and 2 to permutation $(2, 1)$; when $n = 3$, we denote 8:2:2 to be the configuration which assigns 8, 2, 2 heads

---

[5]We use GPUs of 16 GB memory for profiling. BERT with $N = 2048$ fails due to an out-of-memory error.

| $N$ | Model | Training Time (day) | Memory (per GPU, GB) | Heads Config. | Valid. ppl |
|---|---|---|---|---|---|
| 512 | RoBERTa-1seq | 6.62 | 9.73 | - | 3.58 |
| | BlockBERT n=2 | 5.83 (-12.0%) | 7.91 (-18.7%) | 10:2 | 3.56 |
| | BlockBERT n=3 | 5.80 (-12.5%) | 7.32 (-23.8%) | 8:2:2 | 3.71 |
| 1024 | RoBERTa-1seq | 9.66 | 13.39 | - | 3.60 |
| | BlockBERT n=2 | 7.51 (-22.3%) | 9.73 (-27.3%) | 9:3 | 3.57 |
| | BlockBERT n=3 | 7.23 (-25.1%) | 8.55 (-36.1%) | 8:2:2 | 3.63 |

Table 2: Pre-training Performance Analysis.

to permutation $(1, 2, 3)$, $(2, 3, 1)$, and $(3, 1, 2)$, resp. We compare BlockBERT with the following baselines:

**Google BERT** The pre-trained base model from Devlin et al. (2019).

**RoBERTa-2seq** and **RoBERTa-1seq** We compare with two versions of RoBERTa (Liu et al., 2019). RoBERTa-2seq is trained with both masked language model (MLM) task and next sentence prediction (NSP) task, while RoBERTa-1seq refers to the pre-training model with only MLM task.

**SparseBERT** We pre-train BERT models with its Transformer encoder replaced by a Sparse Transformer encoder (Child et al., 2019). We set its sparsity hyper-parameters stride $\ell = 128$ and expressivity $c = 32$. The attention masks used for Sparse Transformer encoder are illustrated in Figure 5.

### 4.1 PRE-TRAINING

All the models follow the base setting, i.e., $L = 12$, $H = 768$, $A = 12$ and are trained on the same corpus — BooksCorpus and English Wikipedia with uncased word piece tokens. We fix the number of tokens per batch $B \times N = 131,072$, i.e., if sequence length $N = 512$ then batch size $B = 256$, if sequence length $N = 1024$ then batch size $B = 128$. The detailed pre-training configuration is listed in Table 6 in Appendix A.1. Moreover, the pre-training of SparseBERT and BlockBERT follows the RoBERTa-1seq setting, i.e., we drop the NSP (Next Sentence Prediction) task, and an input sequence is up to $N$ tokens until it reaches a document boundary. A summary of the pre-training performance comparison between BlockBERT and RoBERTa-1seq is shown in Table 2. Besides memory saving, we also achieve a significant speedup. For example, when $N = 1024$, BlockBERT ($n = 2$) reduces the training time from RoBERTa's 9.7 days to 7.5 days.

### 4.2 FINE-TUNING TASKS

We evaluate BlockBERT on several question answering tasks, including SQuAD 1.1/2.0 (Rajpurkar et al., 2018) and five other tasks from the MrQA shared task[6] — HotpotQA (Yang et al., 2018), NewsQA (Trischler et al., 2017), SearchQA (Dunn et al., 2017), TriviaQA (Joshi et al., 2017) and NaturalQA (Kwiatkowski et al., 2019). Since MrQA does not have an official test set, we follow Joshi et al. (2019a) who split the development set evenly to build a new development set and test set.

These QA datasets have different paragraph length distribution patterns and are thus ideal for testing the effectiveness of BlockBERT. For example, SQuAD, NaturalQA, and HotpotQA consist of mostly short paragraphs (shorter than 512), while paragraphs in SearchQA (average length 1,004) and TriviaQA (average length 934) have around 1,000 tokens. This means that for SearchQA and TriviaQA, a BERT model with sequence length $N = 512$ can only capture half of the context. The detailed paragraph length distributions can be found in Figure 6.

For all the pre-trained models, we adopt the same fine-tuning QA setup from Devlin et al. (2019). The tokenized paragraph $(p_1, \cdots, p_s)$ and question $(q_1, \cdots, q_t)$ are concatenated to be a sequence [CLS]$q_1 \cdots q_t$[SEP]$p_1 \cdots p_s$[SEP]. The sequence is then fed into the pre-trained model with two extra linear layers for predicting the start and end positions of the answer spans. The detailed fine-tuning setting is listed in Appendix A.4. Table 3 and Table 4 report the experimental results.

---

[6]https://mrqa.github.io

| $N$ | Model | SQuAD 1.1 | | SQuAD 2.0 | |
| --- | --- | --- | --- | --- | --- |
| | | EM | F1 | EM | F1 |
| - | Human Perf. | 82.30 | 91.20 | 86.80 | 89.40 |
| 512 | Google BERT | 81.19 | 88.45 | 74.08 | 77.16 |
| | XLNet | - | - | 78.46 | 81.33 |
| | RoBERTa-2seq | 82.91 | 89.78 | 75.79 | 79.17 |
| | RoBERTa-1seq | **84.43** | **91.48** | **79.22** | **82.27** |
| | SparseBERT | 80.49 | 88.09 | 74.15 | 76.96 |
| | BlockBERT n=2, 10:2 | *84.08* | *90.77* | *78.34* | *81.46* |
| | BlockBERT n=3, 8:2:2 | 82.37 | 89.64 | 77.33 | 80.33 |
| 1024 | RoBERTa-1seq | **84.58** | **91.14** | **79.34** | **82.26** |
| | SparseBERT | 81.02 | 88.37 | 74.51 | 77.57 |
| | BlockBERT n=2, 9:3 | *83.65* | *90.74* | *78.55* | *81.45* |
| | BlockBERT n=3, 8:2:2 | 82.74 | 90.05 | 76.79 | 79.84 |

Table 3: Dev set results on SQuAD 1.1/2.0. The result of XLNet(-Base) is from (Yang et al., 2019).

**BlockBERT (n=2) v.s. RoBERTa-1seq** Comparing BlockBERT ($n = 2$) with RoBERTa-1seq on pre-trained model with $N = 512$, we observe an absolute F1 difference from 0.04 (in NaturalQA) to 1.18 (in NewsQA), with average difference to be 0.55. For $N = 1024$, BlockBERT achieves more comparable or even better performance (in SearchQA, NewsQA, and HotpotQA) to RoBERTa-1seq. The average difference on F1 reduces to 0.27.

**BlockBERT v.s. SparseBERT** For $N = 512$, it is interesting that BlockBERT with 3 blocks (density 33.33%) performs better then SparseBERT (density 44.20%) in both SQuAD and MrQA tasks. Similar patterns can be observed for $N = 1024$. These results show that off-diagonal masking matrices, e.g., the masking matrix defined by permutation $(2, 1)$, play crucial roles in BlockBERT.

**Effect of Long Sequence Pre-training** Our observations are twofold. (1) Long sequence pre-training benefits long sequence fine-tuning. In TriviaQA and SearchQA, of which paragraph lengths are around 1024, pre-training models with $N = 1024$ achieve significantly better performance. (2) The heterogeneity of pre-training and fine-tuning sequence length may hurt performance. For example, in SQuAD, we do not see significant performance gain by using pre-trained models with $N = 1024$; in HotpotQA and NewsQA, longer sequence pre-training even hurts performance.

**Effect of #Blocks** It is not surprising that BlockBERT with 2 blocks ($n = 2$) performs better than that with 3 blocks ($n = 3$), because it keeps more attention matrix entries. The biggest difference is in SQuAD 2.0 and NewsQA with $N = 1024$, where we observe an absolute loss of 1.6 F1 by increasing block number from 2 to 3.

In summary, not only BlockBERT saves training time and memory, but it also has competitive and sometimes better performance, especially for tasks with longer sequences. This demonstrates the effectiveness of our blockwise multi-head attention approach.

| $N$ | Model | SearchQA | | TriviaQA | | NewsQA | | NaturalQA | | HotpotQA | |
| --- | --- | --- | --- | --- | --- | --- | --- | --- | --- | --- | --- |
| | | EM | F1 | EM | F1 | EM | F1 | EM | F1 | EM | F1 |
| 512 | Google BERT | 74.94 | 80.37 | 70.18 | 75.35 | 51.27 | 66.25 | 66.13 | 78.29 | 60.50 | 77.08 |
| | RoBERTa-2seq | 76.12 | 81.74 | 71.92 | 76.79 | 52.45 | 66.73 | 66.98 | 78.63 | 61.52 | 77.81 |
| | RoBERTa-1seq | **77.09** | **82.62** | **73.65** | **78.22** | **56.13** | **70.64** | **67.14** | **79.07** | **62.77** | **79.28** |
| | SparseBERT | 73.36 | 79.01 | 68.71 | 73.15 | 51.18 | 65.47 | 65.53 | 77.46 | 58.54 | 74.85 |
| | BlockBERT n=2, 10:2 | *76.68* | *82.33* | *72.36* | *77.53* | *54.66* | *69.46* | *66.94* | *79.03* | *62.13* | *79.15* |
| | BlockBERT n=3, 8:2:2 | 75.54 | 81.07 | 72.05 | 76.74 | 53.82 | 68.39 | 66.14 | 78.47 | 60.64 | 77.46 |
| 1024 | RoBERTa-1seq | 77.47 | 83.12 | **75.29** | **80.20** | 55.00 | 69.64 | **68.28** | **80.35** | 61.89 | 78.71 |
| | SparseBERT | 74.83 | 80.54 | 70.56 | 75.34 | 51.67 | 67.16 | 65.07 | 77.31 | 59.65 | 76.02 |
| | BlockBERT n=2, 9:3 | *77.95* | *83.51* | *75.06* | *79.41* | *55.44* | *70.08* | *67.31* | *79.39* | *62.13* | *78.94* |
| | BlockBERT n=3, 8:2:2 | 76.98 | 82.76 | 74.78 | 79.28 | 53.48 | 68.50 | 65.91 | 78.20 | 61.89 | 78.18 |

Table 4: MrQA test results (Tasks are sorted decreasingly by average paragraph length).

### 4.3 ABLATION STUDY

We fix the assignment of attention heads in above experiments. For example, BlockBERT with sequence length $N = 512$ and 2 blocks is trained with ten heads using permutation $(1, 2)$ and the other two using permutation $(2, 1)$. However, we know that there are other ways to partition twelve attention heads, e.g., seven heads for permutation $(1, 2)$ and the other five for permutation $(2, 1)$. It would be interesting to see how the assignment of heads affects model performance. In this section, we grid search attention head assignments and plot their best validation performance in 1.2M training steps. The results are shown in Figure 4.

Our observations are threefold: (1) Identity permutations, i.e., $(1, 2)$ and $(1, 2, 3)$, are important. As shown in Figure 4, all optimal solutions assign considerable attention heads to block diagonal matrices, since those matrices enable each token to attend to its nearby tokens; (2) Non-identity permutations follow the rule of "vital few and trivial many." Although identity permutations are important, assigning all attention heads to them (corresponding to 12:0 and 12:0:0 in Figure 4) significantly hurts performance, since the model can not learn long-term dependencies with only identity permutation; (3) Pre-training performance and fine-tuning performance are correlated but not always consistent. When $n = 3$, pre-training performance suggests 10:1:1 to be the best head assignment — ten heads for permutation $(1, 2, 3)$, one head for $(2, 3, 1)$ and one head for $(3, 1, 2)$, but we observe that the configuration of 8:2:2 achieves better performance in fine-tuning tasks.

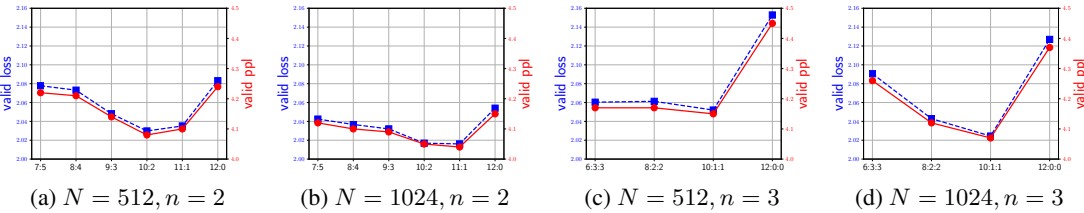

(a) $N = 512, n = 2$   (b) $N = 1024, n = 2$   (c) $N = 512, n = 3$   (d) $N = 1024, n = 3$

Figure 4: Ablation over blockwise attention heads assignment.

## 5 RELATED WORK

In this section, we review the related work of memory optimization for neural network training and recent efforts to simplify Transformer and BERT. In recent years, there is an increasing interest in training neural networks with low-memory (Sohoni et al., 2019). Mainstream techniques include low-precision training (Micikevicius et al., 2017), microbatching (Huang et al., 2018), gradient checkpointing (Chen et al., 2016). Another line of researches studies this problem from a theoretical perspective, including the recently proposed lottery ticket hypothesis (Frankle & Carbin, 2018). Since the invention of Transformer (Vaswani et al., 2017; Dai et al., 2019) and its successful application on language model pre-training (Devlin et al., 2019; Radford et al., 2019; Yang et al., 2019; Liu et al., 2019), there have been several studies attempted to simplify it from different perspectives. The first line of research focuses on attention matrix sparsification, such as Star Transformer (Guo et al., 2019), Sparse Transformer (Child et al., 2019), Adaptive Sparse Transformer (Correia et al., 2019; Sukhbaatar et al., 2019), Log-Sparse Transformer (Li et al., 2019), etc. However, due to limited support for sparse tensor from current deep learning platforms, most of studies have to represent a sparse matrix using a dense matrix with a binary mask or rely on customized CUDA kernels (Gray et al., 2017). The second line of research attempts to prune redundant heads in Transformer, such as Voita et al. (2019) and Michel et al. (2019). The third line of research focuses on knowledge distillation, including DistilBERT[7] which distills BERT using a smaller BERT and Tang et al. (2019) which distills BERT with BiLSTM (Hochreiter & Schmidhuber, 1997).

---

[7]https://github.com/huggingface/pytorch-transformers/tree/master/examples/distillation

# 6 CONCLUSION

In this work, we study lightweight BERT model with the goal of achieving both efficiency and effectiveness. We profile and analyze the memory bottlenecks of BERT, and focus on optimize dot-product self-attention, which consumes quadratic memory with respect to the sequence length. To reduce both training time and memory consumption, we present BlockBERT, which sparsifies the attention matrices to be sparse block matrices. The proposed model achieves time and memory saving without significant loss of performance. In the future, we would like to explore more applications of BlockBERT on NLP tasks involving long sequences such as coreference resolution (Joshi et al., 2019b) and document-level machine translation (Miculicich et al., 2018), and also non-NLP tasks such as protein sequence modeling (Rives et al., 2019).

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

## A  APPENDIX

### A.1  NOTATIONS AND PRE-TRAINING HYPER-PARAMETERS

The notations and pre-training hyper-parameters are listed in Table 5 and Table 6.

| | Description | Base | Large |
|---|---|---|---|
| $B$ | Batch size | 256 | 256 |
| $A$ | # Self-attention heads | 12 | 16 |
| $L$ | # Layers | 12 | 24 |
| $H$ | # Hidden units | 768 | 1024 |
| $4H$ | # Feed-forward hidden units | 3072 | 4096 |
| $N$ | Sequence length | 512 | 512 |

Table 5: BERT notations.

| Hyper-parameter | Value |
|---|---|
| Vocabulary Size | 30,522 |
| Dropout | 0.1 |
| Attention dropout | 0.1 |
| Warmup steps | 10K |
| Weight decay | 0.01 |
| Max steps | 2.4M |
| Initial learning rate | 0.00025 |
| Learning rate decay | Linear |
| Adam $\epsilon$ | 1e-8 |
| Adam $\beta_1$ | 0.9 |
| Adam $\beta_2$ | 0.999 |
| Gradient Clipping | 1.0 |

Table 6: Pre-training hyper-parameters.

### A.2  PROFILER IMPLEMENTATION

Among the three types of training memory, model memory and optimizer memory is relatively easy to profile (can be computed by enumerate each tenor and summing up `tensor.numel() * tensor.element_size()`). To calculate activation memory, Sohoni et al. (2019) traverse PyTorch's autograd graph and sum up necessary storage space. They find that the summation of model memory, optimizer memory and activation memory matches PyTorch memory profiling tool `torch.cuda.max_memory_allocated`. Based on their observation, we use

$$\texttt{torch.cuda.max\_memory\_allocated} - \text{model memory} - \text{optimizer memory} \qquad (4)$$

as an estimate to activation memory. When profiling BERT, we first pre-train it for 1000 steps, and then compute its model and optimizer memory. Finally, we esitmate its activation memory according to Equation 4.

## A.3 SPARSEBERT

The sparse masking matrices we use for Sparse Transformer (Child et al., 2019) are shown in Figure 5. We adopt the implementation from Fairseq[8].

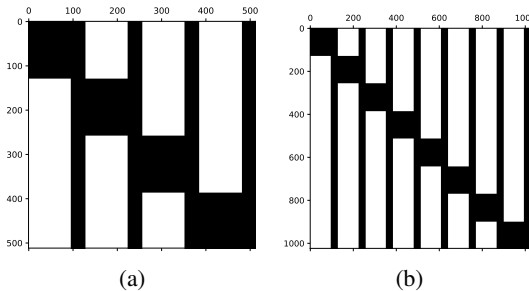

(a)        (b)

Figure 5: The sparse masking matrices we use in Sparse Transformer (fixed mode) encoder. White color indicates attention values to be masked. (a) $N = 512, \ell = 128, c = 32$, density 44.20%; (b) $N = 1024, \ell = 128, c = 32$, density 34.97%.

## A.4 FINE-TUNING SETTINGS

Our fine-tuning is implemented based on code base from HuggingFace[9] and SpanBERT (Joshi et al., 2019a). We use `max_sequence_length`=$N$, i.e., we allow fine-tuning task to input sequences as long as the pre-training model. If the input sequence is too long to fit the `max_sequence_length`=$N$ constraints, we use a sliding window of stride 128 to split it. We grid search learning rate from {5e-6, 1e-5, 2e-5, 3e-5, 5e-5} and batch size from {16, 32}. The fine-tuning is performed for 4 epoches.

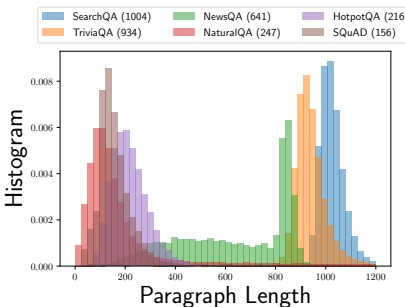

Figure 6: Paragraph length (after tokenization) distribution. The distribution of SQuAD 2.0 is very similar to SQuAD 1.1, so we only plot SQuAD 1.1 here.

## A.5 TEST EFFICIENCY

We benchmark test efficiency of RoBERTa and our proposed BlockBERT. The benchmark code follows huggingface[10]. All experiments are run 30 times on a 32GB V100 GPU with half precision (FP16). We report the average running time at Table 7. As we can see, BlockBERT does achieve speedup and memory reduction during test time. Take 8×1024, i.e., batch size $B = 8$, sequence length $N = 1024$, as an example, we can see that BlockBERT with 2 blocks saves 27.8% of test time, and BlockBERT with 3 blocks saves more (30.4%). As for memory, we can observe that RoBERTa can not handle an input of 16×1024, while it is possible for BlockBERT to work on it.

---

[8]https://github.com/pytorch/fairseq/blob/master/fairseq/modules/sparse_multihead_attention.py.

[9]https://github.com/huggingface/pytorch-transformers

[10]https://github.com/huggingface/transformers/blob/master/examples/benchmarks.py

| $B \times N$ | $8 \times 1024$ | $16 \times 1024$ | $24 \times 1024$ | $32 \times 1024$ |
|---|---|---|---|---|
| RoBERTa | 0.1371 | OOM | OOM | OOM |
| BlockBERT n=2 | 0.0990 | 0.1869 | OOM | OOM |
| BlockBERT n=3 | 0.0954 | 0.1790 | 0.2634 | OOM |

Table 7: Test time statistics (sec) for different input size (Batch size $\times$ Sequence length). OOM indicates out-of-memory.

