# OpenReview forum: "Blockwise Self-Attention for Long Document Understanding"
_ICLR.cc/2020/Conference — Reject_

### Official Review · AnonReviewer2 · 2019-10-23
**Official Blind Review #2**

**Rating:** 6

**Review:**

The paper propose to sparsify the attention matrix to decrease memory usage and to speed up training. The authors experiment the model on multiple tasks. The model gains ~20% efficiency with ~20% decrease in memory use while maintaining comparable performance to the state of the art model. To keep the performance comparable, the authors use the same training corpus. The authors also discuss how block size could change the performance of the model. The paper is clear and well organized with good experiment results.

**Experience Assessment:**

I have read many papers in this area.

**Review Assessment: Checking Correctness Of Derivations And Theory:**

I did not assess the derivations or theory.

**Review Assessment: Checking Correctness Of Experiments:**

I assessed the sensibility of the experiments.

**Review Assessment: Thoroughness In Paper Reading:**

I read the paper at least twice and used my best judgement in assessing the paper.

---

> ### Author Response · Authors · 2019-11-13
> **To AnonReviewer2. Thank you for your positive feedback.**
>
> We appreciate your positive feedback and will keep polishing our work.

---

### Official Review · AnonReviewer1 · 2019-10-25
**Official Blind Review #1**

**Rating:** 3

**Review:**

This paper introduces a optimisation for BERT models based on using block matrices for the attention layers. This allows to reduce the memory footprint and the processing  time during training while reaching state-of-the-art results on 5 datasets. An interesting study on memory consumption in BERT is conducted. No results are given at test time : is there also a memory and processing time reduction ?

Even if the proposition is interesting, the impact of the paper is limited to the (flourishing) scope optimising Bert models ("Bertology"). The authors do not mention if their code is available.


Table 3 : Humam -> Human

**Experience Assessment:**

I do not know much about this area.

**Review Assessment: Checking Correctness Of Derivations And Theory:**

N/A

**Review Assessment: Checking Correctness Of Experiments:**

I assessed the sensibility of the experiments.

**Review Assessment: Thoroughness In Paper Reading:**

I made a quick assessment of this paper.

---

> ### Author Response · Authors · 2019-11-13
> **To AnonReviewer2. Thank you for your feedback and insightful discussions.**
>
> We appreciate your feedback and will keep polishing our work. Thank you very much.
>
> We believe the reviewer has misunderstood the scope of the paper’s contribution. This paper addresses an issue with *self-attention*, which is used (via transformers) in many other applications other than BERT, such as machine translation, speech recognition, and even computer vision. Therefore, our proposed blockwise self-attention is not only applicable to “optimising BERT models (Bertology)” - but can potentially assist any domain that involves modeling long sequences via self-attention.
>
> Regarding the test time efficiency, BlockBERT does achieve speedup and memory reduction during test time. The test-time (sec) statistics is as follows (OOM indicates out-of-memory):
> | Model/Batch x Length | 8 x 1024         | 16 x 1024      | 24 x 1024     | 32 x 1024 |
> | RoBERTa                         | 0.137146792  | OOM             | OOM            | OOM        |
> | BlockBERT (n=2)            | 0.098985166 | 0.186883354 | OOM            | OOM        |
> | BlockBERT (n=3)            | 0.095399417 | 0.179042275 | 0.26340443 | OOM        |
>
> Take 8x1024 (Batch size=8, Sequence Length=1024) as an example, we can see that BlockBERT with 2 blocks saves 27.8% of test time, and BlockBERT with 3 blocks saves more (30.4%). As for memory, we can observe that RoBERTa can not handle an input of 16x1024, while it is possible for BlockBERT to work on it.
>
> All experiments are run 30 times on a 32GB V100 GPU with half precision (FP16). We report the average running time. We have added detailed test efficiency analysis at A.5.
>
> As for the code, we have released the code publicly and will include the link in the paper once deanonymized.
>
> Thanks for pointing out the typo. We have fixed it.

---

### Official Review · AnonReviewer4 · 2019-10-26
**Official Blind Review #4**

**Rating:** 6

**Review:**

The authors propose BlockBERT, a model that makes the attention matrix of Transformer models sparse by introducing block structure. This has the dual benefit of reducing memory and reducing training time. The authors show on various question answering tasks that their model is competitive with RoBERTa.

1. Can the authors add additional details about the training of their model in Section 4.1? It is not clear for example the vocabulary size - is that the RoBERTa vocabulary size of around 50k or the BERT vocabulary size that is smaller? I believe this will affect the memory and training speed.

2. For the datasets such as TriviaQA and SearchQA, how is RoBERTa finetuned on these tasks? By doing the window approach?

3. The authors compare to RoBERTa and Sparse BERT as baselines for the section on performance. However, can the authors also include metrics on training time and memory in Table 2 for Sparse BERT as well as other sparse attention transformer architectures proposed (for example the Correia paper or the Sukhbaatar paper)? It is not clear the savings from this architecture compared to sparse Transformers in general.

4. The analysis in Section 4.3 is quite unclear due to how compressed the description is and how tiny the graphs are.

5. The authors mention that attention heads can be sparsified due to the memory usage and quadratic time complexity. Other work has also shown that the attention heads are quite redundant and can be pruned away, so attention head dropout is effective. For example https://arxiv.org/abs/1905.09418


**Experience Assessment:**

I have published one or two papers in this area.

**Review Assessment: Checking Correctness Of Derivations And Theory:**

I assessed the sensibility of the derivations and theory.

**Review Assessment: Checking Correctness Of Experiments:**

I assessed the sensibility of the experiments.

**Review Assessment: Thoroughness In Paper Reading:**

I read the paper at least twice and used my best judgement in assessing the paper.

---

> ### Author Response · Authors · 2019-11-13
> **To AnonReviewer4. Thank you for your positive feedback.**
>
> Thank you for your comments. We updated the paper to add detailed experimental setting and related work, and also polish the presentation according to your feedback. Here are the replies to your comments.
>
> 1.Vocab size.
> We pretrained our RoBERTa/BlockBERT/SparseBERT with the same vocabulary as Google’s BERT (uncased version). All of them are of vocabulary size 30,522.
>
> 2. Fine-tuning settings.
> We allow fine-tuning task to input sequences as long as the pre-training model, i.e., if the pre-trained model has max_sequence_length=1024, the fine-tuning task can feed a 1024 token sequence to it. If the input sequence is too long to fit the max sequence length constraints, we use a sliding window of stride 128 to split it. We grid search learning rate from {5e-6, 1e-5, 2e-5, 3e-5, 5e-5} and batch size from {16, 32}. The fine-tuning is performed for 4 epoches. We update detailed fine-tuning settings at Appendix A.1.
>
> 3. Training time and memory of SparseBERT.
> We implement SparseBERT in a direct way, with the goal of comparing performance, not speed. We first compute the N^2 attention matrix, and then mask it to be a sparse matrix according to the sparse pattern defined in Sparse Transformer paper. Consequently, this implementation of SparseBERT has very close training time/memory cost as RoBERTa (as it can not avoid the O(N^2) attention computation). We did so because the code released by Sparse Transformer is based on Tensorflow and relies on customized CUDA kernels, but our pre-training is done using PyTorch.
>
> 4. About Section 4.3
> In section 4.3, we mainly discuss how the distribution of attention heads affects the model performance. The main conclusion is that we need far more attention heads for identity permutation (which learns local dependencies) than non-identity permutations (which learns long-term dependencies). We have polished the presentation according to your feedback.
>
> 5. Redundancy in attention heads
> Thanks for the reference. And yes, our method can be viewed as a way to compress (or factorize) redundant attention heads. We have added it to reference and discuss it in related work.

---

### Author Response · Authors · 2019-11-13
**To All Reviewers. Revision Log.**

Dear reviewers, we have updated our submission according to your helpful comments. We made the following changes:

1. We added test efficiency analysis at Appendix A.5. According to our analysis, we can see that BlockBERT achieves memory and time saving in both training and testing.

2. We polished the presentation in section 4.3 to make the ablation study section more readable.

3. We added detailed pre-training and fine-tuning setting in Appendix A.1 and A.4, respectively.

4. We added some references as suggested by reviewers in related work section (Section 5).

5. We fixed some typos.

Thank you very much.

---

### Decision · Program_Chairs · 2019-12-19

**Decision:**

Reject

**Comment:**

This paper proposes blockwise masked attention mechanisms to sparsify Transformer architectures, the main motivation being reducing the memory usage with long sequence inputs. The resulting model is called BlockBERT. The paper falls in a trend of recent papers compressing/sparsifying/distilling Transformer architectures, a very relevant area of research given the daunting resources needed to train these models.

While the proposed contribution is very simple and interesting, it also looks a rather small increment over prior work, namely Sparse Transformer and Adaptive Span Transformer, among others. Experiments are rather limited and the memory/time reduction is not overwhelming (18.7-36.1% less memory, 12.0-25.1% less time), while final accuracy is sometimes sacrificed by a few points. No comparison to other adaptively sparse attention transformer architectures (Correia et al. EMNLP 19 or Sukhbaatar et al. ACL 19) which should as well provide memory reductions due to the sparsity of the gradients, which require less activations to be cached. I suggest addressing this concerns in an eventual resubmission of the paper.